# False Positive Decremented Research for Fire and Smoke Detection in Surveillance Camera using Spatial and Temporal Features Based on Deep Learning

**Yeunghak Lee [1] and Jaechang Shim [2],***

[1] Department of Multimedia Engineering, Andong National University, Andong 36729, Korea; yhyi@anu.ac.kr
[2] Department of Computer Engineering, Andong National University, Andong 36729, Korea
* Correspondence: jcshim@anu.ac.kr; Tel.: +82-10-9770-5645

**Abstract:** Fire must be extinguished early, as it leads to economic losses and losses of precious lives. Vision-based methods have many difficulties in algorithm research due to the atypical nature fire flame and smoke. In this study, we introduce a novel smoke detection algorithm that reduces false positive detection using spatial and temporal features based on deep learning from factory installed surveillance cameras. First, we calculated the global frame similarity and mean square error (MSE) to detect the moving of fire flame and smoke from input surveillance cameras. Second, we extracted the fire flame and smoke candidate area using the deep learning algorithm (Faster Region-based Convolutional Network (R-CNN)). Third, the final fire flame and smoke area was decided by local spatial and temporal information: frame difference, color, similarity, wavelet transform, coefficient of variation, and MSE. This research proposed a new algorithm using global and local frame features, which is well presented object information to reduce false positive based on the deep learning method. Experimental results show that the false positive detection of the proposed algorithm was reduced to about 99.9% in maintaining the smoke and fire detection performance. It was confirmed that the proposed method has excellent false detection performance.

**Keywords:** deep learning; fire and smoke detection; spatial and temporal; wavelet transform; coefficient of variation

## 1. Introduction

Many civilian fire injuries and civilian fire deaths occur each year due to intentionally-set fires and naturally occurring fires, which causes much property damage. Fires are classified into structure fires (home structures which include one-and two-family, manufactured homes, and apartments), non-residential structure fires (public assembly, school and college, store and office, industrial facilities, and other structures), and outdoor fires (bush, grass, forest, rubbish, and vehicle fires) [1]. Research on automatic fire detection or monitoring has long been the focus of the interior structure fires and non-residential structure fires to protect casualties and property damage from fires.

Smoke is very important because it indicates the start of a fire. However, sometimes the flames start first; thus, both smoke and flames require early detection to extinguish the fire early. Many methods of detecting smoke and flames to extinguish a fire early have been studied. In order to reduce the damage caused by fire, many early fire detection systems using heat sensors, smoke sensors, and flame detection sensors that detect flames by infrared rays (spectrum) and ultraviolet rays (spectrum) are frequently used [2,3]. Sensors used in buildings, factories, and interior spaces detect the presence of particles produced by fire flames and smoke in close proximity using a chemical reaction by ionization

that requires proximity. Traditional fire alarm systems using sensors show good detection results in close proximity for activation or very narrow spaces [4,5]. However, sensor-based sensing systems are expensive because many devices need to be installed for fast detection. The disadvantage of the thermal sensor is that the detection is slow because it uses the temperature difference from the surroundings. The smoke sensor may be delayed depending on the speed of the smoke or may not be detected depending on the air flow. In addition, sensor-based detection systems cannot provide users with information about the location or size of a fire. The main disadvantage of the sensor based system is that it is difficult to install outdoors. As mentioned earlier, fires can occur anywhere and anytime, and must be detected at various locations.

In order to overcome the shortcomings of the sensor-based detection systems, many methods of detecting smoke and fire using camera sensors (image-based) have been studied [6,7]. Compared to sensor based fire detectors, video fire detectors have many advantages, such as fast response, long range detection and large protected areas. However, most of the recent video fire detections have a high rate of false alarms [8].

Vision based fire detection includes short range fire detection and long range fire detection. Long-distance forest or wildfire smoke and fire detection system using fixed CCD (Charge-Coupled Device) cameras is the monitoring of smoke and fire from distant mountains or fields [9–11]. In addition, Zhao et al. [12] described wildfire identification based on deep learning using unmanned aerial equipment. To extract local extremal regions of smoke, they used the rapidly growing Maximally Stable Extremal Region detection method in the field of initial smoke region detection.

More research has been conducted on short distance fire and smoke detection than on long distance forest of wildfire. Early fire detection using cameras detected fires in tunnels and mountains using black and white images [13,14]. Early feature extraction detected flames by measuring histogram changes using the temporal change characteristics of flames from black and white images. Recently, image-based flame detection methods using motion, color, shape, texture, and frequency analysis have been studied for the last 20 years [15–21].

Conventional flame detection methods include a method using RGB (Red, Green, Blue) HSV (Hue, Saturation, Value), YCbCr color models, etc., wavelet transform after detecting moving areas and flame color pixels, flame intensity changes over time, the shape of contour of fire flame in HSV color models and time-space domain, and a method using infrared image.

The color image fire detection algorithms determine cases where the flame's color level exceeded a certain threshold in the brightness information of the color space such as RGB, YCbCr, HIS (Hue, Saturation, Intensity), and CIEL * a * b * (CIELAB, Commission Interationale De L'éclairage) [22–26]. Algorithms using spatial domain analysis are algorithms for distinguishing between the flame color and the non-flame color. There are algorithms for determining the fire or analyzing the frequency components of the flame region by analyzing the texture of the flame candidate area [15,26,27]. The algorithm using the frequency analysis of the time domain determines the fire by analyzing the frequency of a specific level value of the flame candidate region that changes over time [22,23,25,26].

Chen et al. [27] studied the fire detection system using RGB and HSI color model and rule-based by using the characteristic that the flame movement is spread in irregular shape when fire occurs. Toreyin et al. [28] proposed a system that detects fire and non-fire using temporal and spatial wavelet analysis of input images as a feature of high frequency components, based on the fact that smoke appears translucent in the early stages of fire. Yuan [7] proposed an algorithm which is fast estimated the motion orientation of smoke and an accumulative motion model which is used the integral image. This is a method of generating a direction histogram for a motion vector by using a feature of upward moving of smoke, and determining that smoke is a case when there are a lot of motion vectors in a relatively upward direction. Yuan [29] proposed a smoke detection algorithm based neural network classification to train using feature vectors, which are generated by LPB (Local Binary pattern) and LBPV (Local Binary pattern Variance) histograms for rotation and lighting in multi-scale pyramid images. Celik and Demirel [30] presented the experimental results using YCbCr color space and

proposed a pixel classification algorithm for flames. To this end, they suggested a very innovative algorithm that separates the chrominance from luminance components. However, this method used heuristic membership and did not produce good results for the new data. Fujiwara [31] proposed a smoke detection algorithm for smoke shapes using a fractal encoding method using the self-organism of smoke in grayscale images. Liu and Ahuja [16] detected the fire region based on the area expansion method using the fire initial region that has high brightness. They asserted that the fire zones and non-fire zones are classified by Fourier coefficients change over time. Philips [32] classified the fire region using the changes in status over time for candidate region, after the fire flame candidate region is dedicated by the color histogram adapted Gaussian filter. Tian et al. [33] detected smoke regions by image separation. After the background model was created, the smoke was detected by gray color and partial transparency. The limitation of the vision-based method is that it fails to detect transparent smokes. Moreover, it often mistakenly detects many natural objects, for example, the sun, various artificial lights or light reflects on various surfaces, dust particles, as well as flame and smoke. Additionally, scene complexity and low-video quality can affect the robustness of vision-based flame detection algorithms, thus increasing the false alarm rate. Barmpoutis et al. [34,35] also asserted that high false alarm rates are caused by natural objects, which have similar characteristics with flame, and by the variation of flame appearance. Other causes have claimed environmental changes that complicate fire detection including clouds, movement of rigid body objects in the scene, and sun and light reflections. Hence, the difficulty of fire flame detection from digital images is due to the chaotic and complex nature of fire phenomena. Lee et al. [36] proposed smoke detection algorithm based on the Histogram of Oriented Gradients and LBP. Adaboost, which is constructing a strong classifier as linear combination, was used to classify trained object.

In contrast, the deep learning based fire flame and smoke detection systems have automatic feature extraction; thus, making the process much more reliable and efficient than the conventional feature extraction methods. However, such a deep learning approach requires tremendous computational power, not only during training periods, but also when deploying trained models to hardware to perform specific tasks. As a fire detection method using a security surveillance camera, fire detection techniques using real-time image analysis and deep learning have been proposed.

Recently, several kinds of deep learning algorithms for fire flame and smoke detection have been proposed. Frizzi et al. [37] researched the Convolution Neural Network (CNN) based smoke and flame detection, Sang [38] studied the classification of smoke image and flame image feature using composite product neural network, Wu et al. [39] Studied the detection of fire and smoke regions by extracting dynamic and static features using ViBe algorithm, Shen et al. [40] detected the fire flame using the YOLO (You Look Only Once) model, and Khan et al. [41] also researched a disaster management system to respond to early fire detection and automatic reaction within the inside and outside environment using CNN. Zhang et al. [42] researched forest fire detection utilizing fire patches detection using two joined deep CNNs to detect fire in forest images. However, these models have many parameters to render, which require a large computing space. Thus, these models are unsuitable for onfield fire detection applications using low-cost low-performance hardware. Muhammad et al. [43] used Foggia's dataset [44]. They fine-tuned various variants of CNNs: AlexNet [41], SqueezeNet [43], GoogleNet [44], and MobileNetV2 [45]. They used Foggia's dataset [46] as the major portion of their train dataset. Although Foggia's dataset includes 14 fire and 17 non-fire videos with multiple frames, the dataset contains a lot of similar images, which restricts the performance of the model trained on this dataset to a very specific range of images. Recently, much research has been conducted on Faster R-CNN, which shows higher performance than other network models, like as R-CNN (Region-based Convolutional Network) and Fast R-CNN. Barmpoutis et al. [39] studied higher-order linear dynamical systems based multidimensional texture analysis as the deep learning networks. They classified the fire using the Faster R-CNN model based on the spatial analysis on Grassmann manifold. Wildland forest fire and smoke detection algorithm with Faster R-CNN was suggested by the Zhang et al. [47] to avoid the complex process.

As mentioned above, malfunctions of smoke and flame detection using image processing have been drastically reduced due to the development of deep learning, but the malfunction still exist due to problems of deep learning. The goal of most of the existing approaches is detecting either smoke or fire from images, but as explained, they suffer from a variety of limitations. To solve the problem of these limitations, in this paper, Faster R-CNN model is proposed with object attribution for increasing the smoke and fire flame detection and decreasing the false positive rate. This method is capable of detecting both smoke and fire flame images at the same time, and offers many advantages and exhibits better performance than other existing visual recognition CNN models for the recognition of fire flame and smoke in images. Additionally, we researched a novel algorithm on rigid change of natural environment to reduce the false positive smoke detection based on advanced deep learning, as shown Figure 1.

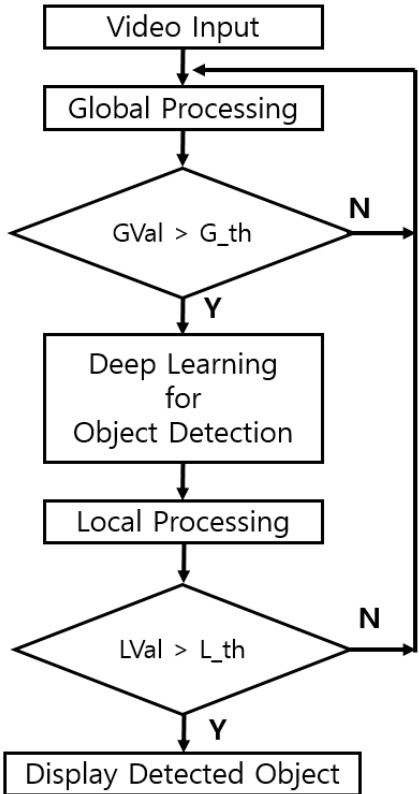

**Figure 1.** Flowchart of the proposed algorithm.

This paper is organized as follows. We propose deep learning model architecture for flame and smoke detection in surveillance camera in Section 2. This paper explains several theories to reduce the rate of false alarms and improve the detection rate in Section 3. Our experimental results and discussion are implemented in Section 4. Finally, the manuscript presents a brief conclusion and future research directions in Section 5.

## 2. Deep Learning (Faster R-CNN)

It is often more difficult to distinguish objects within an image than to classify images. Deep learning using the R-CNN method takes several steps. Once the R-CNN creates a region proposal or a bounding box for an area where an object exists, it unifies the size of the extracted bounding box to use as input to CNN. Next, the model uses SVM (Support Vector Machine) to classify the selected region. Finally, it uses a linear regression model so that the bounding box of the categorized object sets the exact coordinates. CNN for training data is divided into three parts. Figure 2 depicts the full flow of the proposed system. In Figure 2, RPN (Region Proposal network) was used to find a predefined number

of regions (bounding boxes) that can contain objects using features computed by CNN. The next step is to get a list of possible related objects and their locations in the original image. We apply region of interest pooling (ROIP), using boundary boxes for features and related objects extracted from CNN, and extract the features corresponding to related objects as new tensors. Finally, this information is used to classify the contents of the bounding box and the bounding box coordinates are adjusted in the R-CNN module. As a result of the Faster R-CNN, a bounding box of related objects is displayed on the screen. The proposed algorithm part is added at end of Faster R-CNN. We finally select the case where FD (Final Decision) is greater than threshold (TH) using several features in the bounding box.

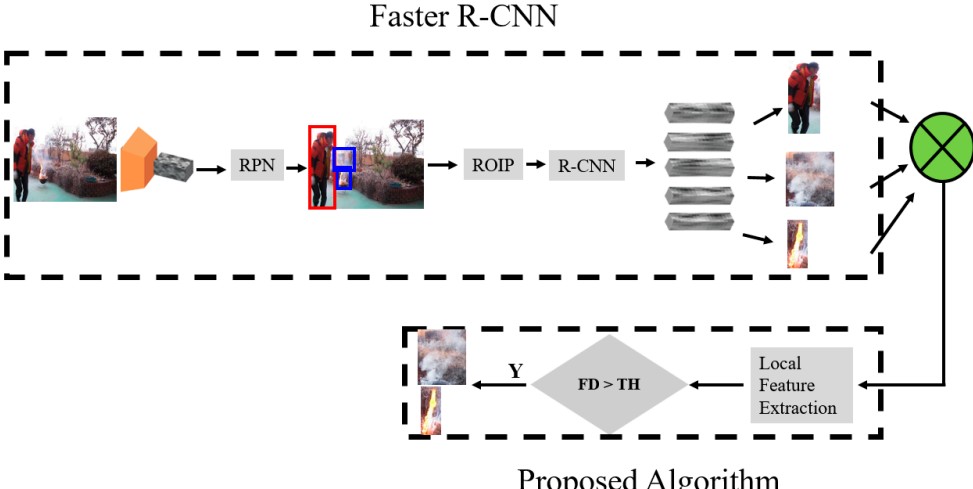

**Figure 2.** Faster R-CNN system flow.

### 2.1. Labeling Dataset

Labeling of the fire flame and smoke in the images was done using the LabelImg program. This paper used a variety size of labeling including fire flame and smoke to train the images, as shown in Figure 3. The labeling results are stored in the .xml file with the image file name along with the four-point coordinates of each rectangle. For labeling dataset, there are two things to be considered. First, a list of class is necessary for the dataset. Second, bounding boxes (Xmin, Ymin, Xmax, Ymax) will be generated by the labeling program according to the classes for images.

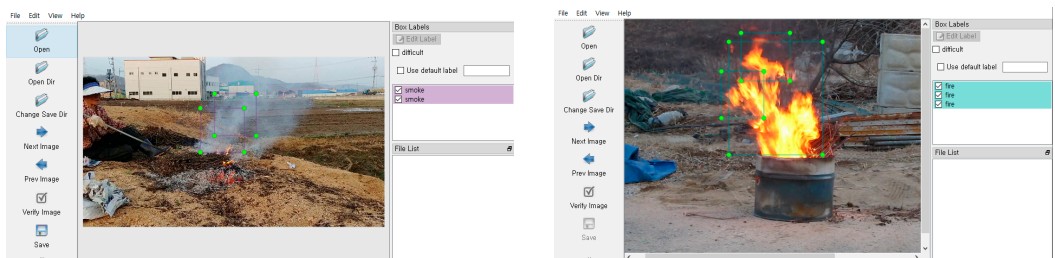

**Figure 3.** Example of labeling area for fire and smoke dataset image.

### 2.2. Training Data with Faster R-CNN

Faster R-CNN [48] is a method of applying a new method called Region Proposal Network (RPN) that merely integrates the part that generates the region proposal within the model. This is a new application of the RPN network for object detection. The function of RPN is to output the rectangle and object score of the part that proposes the object in the input image. It is a fully connected network and is designed to share a convolutional layer with Faster R-CNN. Trained RPN improves the quality of the proposed area and improves the accuracy of object detection. In general, Faster R-CNN searches

external slow selections by CPU calculations but speeds them up by using internal fast RPNs by GPU calculations. The RPN comes after the last convolutional layer, followed by ROIP, classification, and bounding boxes are located, as shown in Figure 4. RPN extracts 256 or 512 features from the input image by convolution calculation using $3 \times 3$ window. This is then used as a box classifier layer and a box regress layer. The predefined reference box name used as the bounding box candidate at each position of the sliding window is used as the box regression. It extracts features by applying predefined anchor boxes of various ratios/sizes using the center position, moving the sliding window of the same size. In our model, we used nine anchor boxes (three sizes and three proportions), and each box is considered as a candidate for the bounding box at each position of the sliding window in the image.

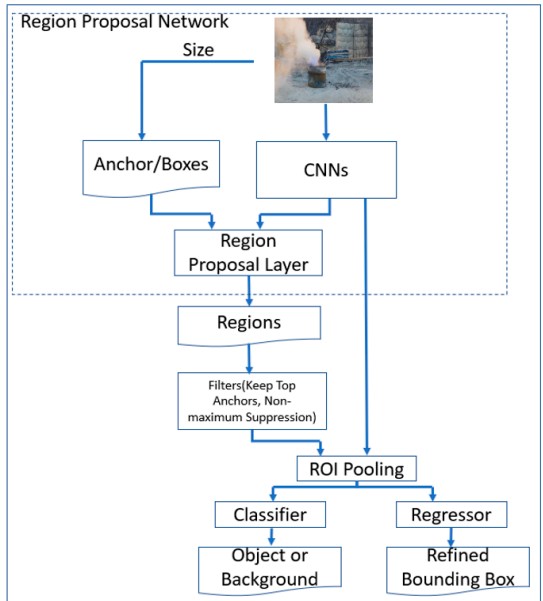

**Figure 4.** The architecture of faster R-CNN.

### 2.3. Creating Inference Graph

An inference graph is also known as a freezing model that is saved for further process. While training the dataset with the model, each pair at different time steps, one is holding the weights ".data", and another is holding the graph ".meta". The labeled image information is progressed using the Faster R-CNN model described above, and the ".meta" file is generated as a training result. The next step is making the graph file (".pb file") which is using the ".meta" file generated in the previous step. Finally, when we use the ".pb" file to detect the objects in the images, the result image including the bounding box and object score will be displayed on the monitor.

## 3. Feature Extraction Methods

### 3.1. Structural Similarity

SSIM (Structural Similarity) [49] is a measure of the similarity of the original image and distortion due to compression and transformation. This is more widely used in signal processing because it has higher accuracy than the Mean Square Error (MSE) method, which uses a measure of the difference between pixel values of two images. We used the evaluation of the test image (X) against the original image (Y) to measure the quantification of visual similarity. The more similar the test image to the

original image, the closer the value is to 1.0, and the more different the test image is to the original image, the closer the value is to 0.0. The SSIM formulas are defined as follows:

$$L(x,y) = \frac{2\mu_x\mu_y + K1}{\mu_x^2 + \mu_y^2 + K1} \tag{1}$$

$$(x,y) = \frac{2\sigma_x\sigma_y + K2}{\sigma_x^2\sigma_y^2 + K2} \tag{2}$$

$$N(x,y) = \frac{\sigma_{xy} + K3}{\sigma_x\sigma_y + K3} \tag{3}$$

where $\mu_x$ and $\mu_y$ are the mean of the pixels, $\sigma_x$ and $\sigma_y$ are the standard deviations, and $\sigma_{xy}$ is covariance. *K1*, *K2*, and *K3* are constants for preventing the denominator and numerator from becoming zero. *L(x, y)* is the relationship of the brightness difference, *M(x, y)* is the contrast difference, and *N(x, y)* is the similarity of the structural change between x and y. The structural similarity is shown in Equation (4):

$$\text{SSIM} = [L(x,y)]^\alpha [M(x,y)]^\beta [N(x,y)]^\gamma \tag{4}$$

where $\alpha$, $\beta$, and $\gamma$ represent the importance of each term; 1.0 was used in this paper.

### 3.2. RGB Color Histogram

Generally, smoke is grayish (dark gray, gray, light gray, and white). Black smoke occurs by unburned materials or a combustion at high temperatures; this means that a certain time has passed since the fire occurred. This paper focuses on the smoke of the initial generation, and sets the conditions as shown in Equation (5) to use smoke colors ranging from gray to white:

$$C = (R + G + B)/3, \quad \tau1 < C_L < \tau2, \quad \tau3 < C_H < \tau4 \tag{5}$$

where *C* is the output image, R is the red image, G is the green image, and B is the blue image. This research set the $C_L$ to a minimum value between 80 ($\tau1$) and 150 and the $C_H$ ($\tau2$) an upper range value between 180 ($\tau3$) and 250 ($\tau4$). The average image *C* is histogrammed into 256 bins (0 to 255) for each pixel. The values stored in each bin of the histogram are normalized using the input image size, and the sum is obtained, as in Equation (6):

$$H_S = \sum_{i=0}^{255} \frac{b_i}{(h \times w)} \tag{6}$$

where $H_S$ is the RGB color histogram result value, $b_i$ means the histogram bins from 0 to 255, which is only included Equation (5) range, and $h$ and $w$ is height and width for an input image. The grayish color is distributed intensively between 80 and 250.

Fire flames are usually bright orange or red (red -> orange -> yellow -> white -> mellow). This paper used HSV color instead of RGB color. The range of HSV color used in the paper is as follows:

- H: 0 to 40
- S: 100 to 255
- V: 80 to 255

As shown in smoke color extraction, HSV color image is also calculated for the average value for the filtered range image. The HSV histogram is obtained by Equation (6).

### 3.3. Coefficient of Variation (CV)

The coefficient of variation is a type of statistic that represents the precision or scatter of a sample, such as variance and standard deviation, in that it shows how scattered the distribution is relative to the mean. CV is a measure of how large the standard deviation is relative to the mean. These coefficients of variation are useful for comparing the spread in two types of data and for comparing variability when the differences between data are large. It is also used to determine the volatility of economic models and securities of economists and investors, as well as areas such as engineering or physics, when conducting quality assurance research and ANOVA gauge R & R [50].

The coefficient of variation is the standard value divided by the mean, as shown Equation (7):

$$CV = \sigma/m \tag{7}$$

where σ is standard deviation and *m* is mean. It showed that the image with smoke and fire flame region has lower *CV* value. In the contrast, the region with false alarm showed higher *CV* value, as shown Figure 5. This paper adapted as the coefficient value (weighting value) of wavelet transform to remove the false alarm cases. In case of Fire flame, we used the R color in RGB color space, and adapted Y color in YCbCr color space for the smoke region.

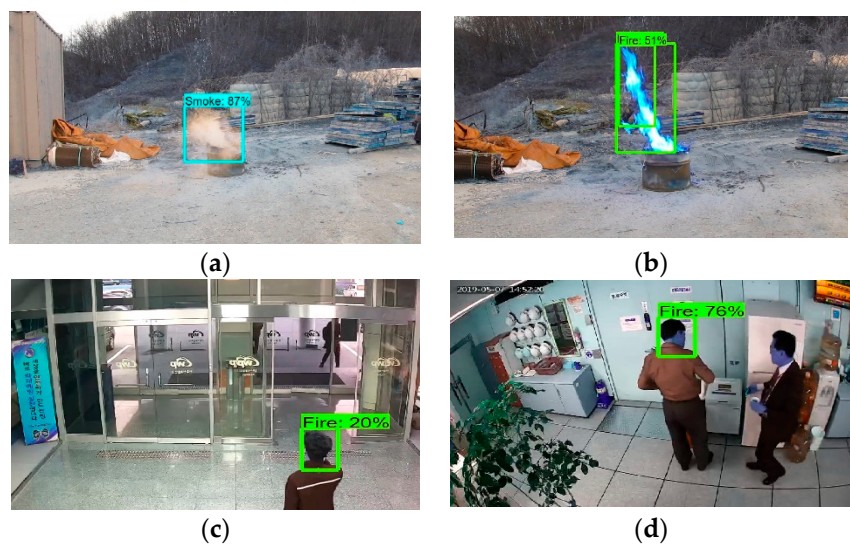

**Figure 5.** The result of coefficient variation values for detected area, (**a**) smoke area Coefficient of Variation (CV) value: 1.5 (87%), (**b**) fire area CV value: 1.9 (51%), (**c**) false alarm area CV value: 6.2 (20%), and (**d**) false alarm area CV value: 13.6 (76%).

### 3.4. Wavelet Transform

In general, smoke is blurry and uneven, thus, it is difficult to detect the contour using the contour detection method. DWT (Discrete Wavelet Transform) [51,52] supports multiple resolutions, and can express contour information of vertical, horizontal, and diagonal components, respectively. Using this feature to represent smoke in DWT energy, it is more apparent than in conventional edge detection methods.

When smoke with translucent characteristics occurs, the smoke part of the image frame is less sharp and the high frequency component is reduced in the area. Wavelet algorithms are generally suitable for expressing image textures and edge characteristics of smoke and fire flames. Background images generally have lower wavelet energy and few moving objects. In contrast, the edge of smoke images becomes less visible, and may disappear from the scene after a certain time. It means that the high frequency energy of the background scene is decreasing. In order to identify smoke in a scene,

any decrease in high frequency from the detected blob images in the frame was monitored by a spatial wavelet transform algorithm.

As shown in Figure 6, if the smoke spreads to the edges of the image, it may be difficult to see initially and the smoke may darken over time, causing part of the background to disappear. [53,54]. This means that there is a high probability that smoke will be present and smoke detection will be easier, as shown in Figure 6. Therefore, this paper used the spatial energy to evaluate the sub-image energy by dividing the image into first stage wavelet transform and summing the squared from each coefficient images in Equation (8):

$$\mathrm{E}(\mathrm{x},\mathrm{y}) = \sqrt{\left[LH(x,y)^2 + HL(x,y)^2 + HH(x,y)^2\right]} \tag{8}$$

where $x$ and $y$ represent positions within the image, and *LH*, *HL*, and *HH* each contain contour information of the high frequency component of the DWT (Discrete Wavelet Transform). *LH* is horizontal low-band vertical high-band, *HL* is horizontal high-band vertical low-band, and *HH* is horizontal high-band vertical high-band. E(x, y) is wavelet energy at each pixel in the candidate region which is detected by deep learning algorithm within each frame.

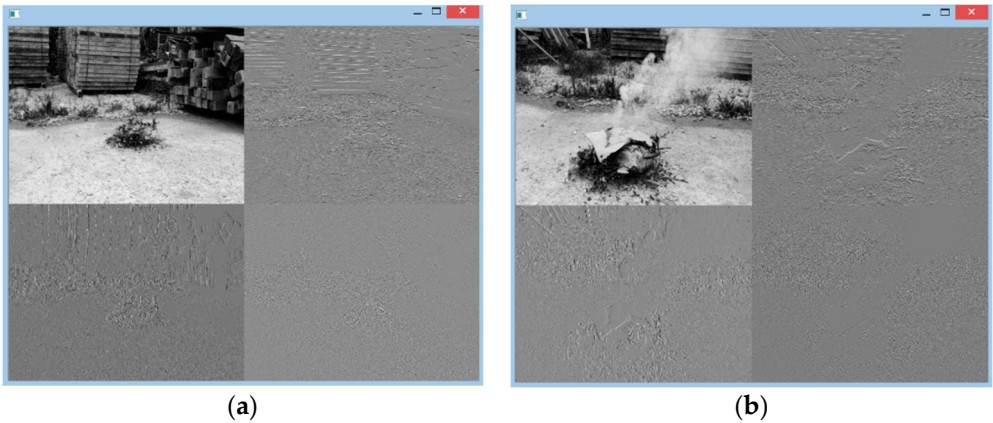

(**a**)　　　　　　　　　　　　　　　　　　　　　　　　(**b**)

**Figure 6.** Single level of wavelet transform results, (**a**) non-smoke sub-images and (**b**) smoke sub-images.

## 4. Experimental Results

We proposed a new algorithm using similarity and color histogram of global and local area in the frame to reduce smoke false positive rate generated by fire detection systems using Onvif camera based on deep learning. In this paper, we used a computer with an Intel Core i7-7700 (3.5 GHz) CPU, 16 GB of memory, and Geforce TITAN-X to perform the experiment. The flame and smoke databases used in this study was obtained from the internet, and general direct ground and factory recorded video. The video recording device was a mobile phone camera, a Canon G5 camera, and a Raspberry pi camera. Python 3.5, Tensorflow, and Opencv were used in this paper.

In order to implement the proposed algorithm, the following process was carried out. The first step is labeling dataset from training database. The first task is labeling data using the LabelImg program, as shown Figure 4. The labeling categories used in this paper are flame, smoke, Grinder, Welding, and human. The result of labeling data is stored in an .xml file that contains the object type name and the four-point coordinates of the object area.

The second step is training process with labeled images. In the training process, the input image is a JPEG or PNG file. The .xml file should be converted to the learning data format of the Tensorflow. Since the meta data and labels of these images are stored in a separate file and must be read separately from the meta data and label file, the code becomes complicated when reading the training data. Additionally, performance degradation can occur if the image is read in JPEG or PNG format and decoded each time. However, the TFRecord file format avoids the above performance degradation and

makes it easier to develop. The TFRecord file format stores the height and width of the image, the file name, the encoding format, the image binary, and the rectangle coordinate of the object in the image. Through this process, the entire training data is classified and stored as 70% training data and 30% validation data. We used the FASTER-CNN ResNet (Deep Residual Network) as the primary model for training, and it is characterized by the smallest number of objects and the highest detection rate. The fire images used in the training consisted of 21,230 pieces.

Finally, we extracted the training model. The learning process stores a check pointer that represents the learning result for the predetermined pointer. Each check pointer has meta information about the Tensorflow model file format and can be learned again. However, because there is a lot of unnecessary information in the ".meta" file, the .meta file needs to be improved to use the actual model. Finally, a ".pb" file is generated that combines the weights except for the unnecessary data in the ".meta" file.

In this paper, we used factory recorded video images, mobile camera, Raspberry pi camera, and general camera as the experimental data. Figure 7 shows an example of continuous frames of video used in the experiment. Fire detection experiment was performed using ".pb" file based on Fater R-CNN model. Figure 8 shows fire and smoke detection results included true positive and false positive using general deep learning.

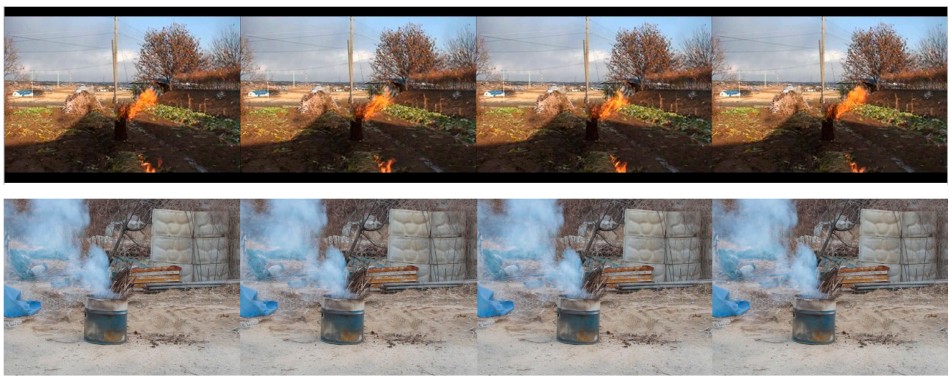

**Figure 7.** Example of the frame sequence of test video.

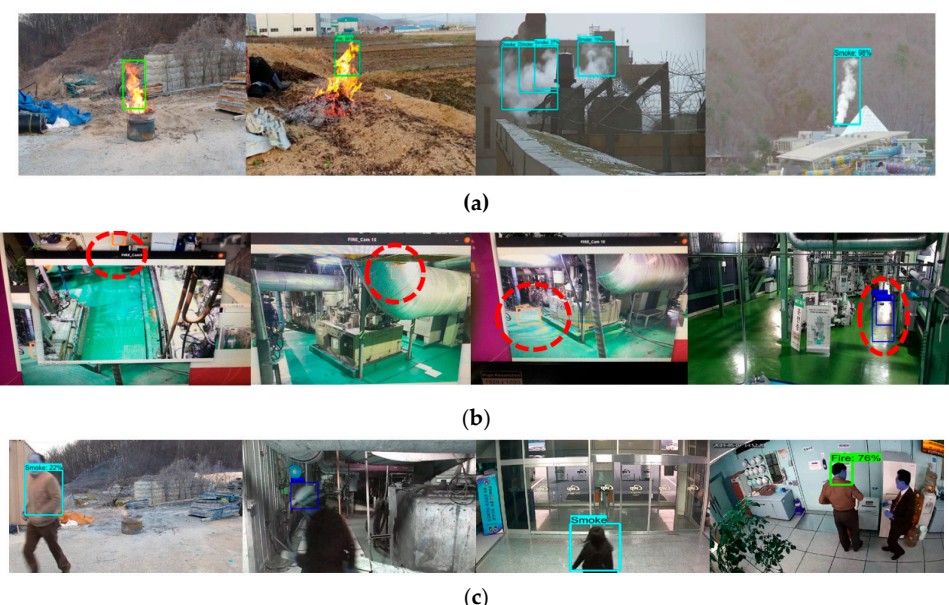

**(a)**

**(b)**

**(c)**

**Figure 8.** The experimental results using the Faster R-CNN: (**a**) the results of true positive, (**b**) the results of false positive (similar shape and color and reflection of sun and light), (**c**) the results of false positive (moving objects and similar color).

Figure 8a shows the result of the experiment to detect fire and smoke using various videos. The detection threshold of Faster R-CNN was 30% or higher. Figure 8b,c shows the result of false positive detection by applying deep learning training results. Although false positives have appeared in many places, there are two types of false positives. First, smoke or flame is detected by reflection of sunlight. Second, facilities inside and outside the factory show similar shapes and colors like smoke and fire. Third, when objects are moving around, deep learning system recognize them as fire flame or smoke for the similar shape of trained fire flame and smoke, as shown in Figure 8c. Table 1 shows the fire and smoke detection results for several videos.

**Table 1.** The results of video test using general Faster R-CNN (frame).

| Videos | Ground Truth | True Positive | True Negative | False Positive |
| --- | --- | --- | --- | --- |
| Video 1 (F/S) | 85 | 85 | 0 | 0 |
| Video 2 (F/S) | 102 | 102 | 0 | 0 |
| Video 3 (F/S) | 890 | 890 | 0 | 890 |
| Video 4 (F/S) | 1159 | 1159 | 0 | 0 |
| Video 5 (F/S) | 1477 | 1477 | 0 | 0 |
| Video 6 (F/S) | 1112 | 1112 | 0 | 0 |
| Video 7 (F/S) | 544 | 544 | 0 | 7 |
| Video 8 (F/S) | 1940 | 1940 | 0 | 0 |
| Video 9 (NON) | 12112 | 0 | 11984 | 128 |
| Video 10 (NON) | 15015 | 0 | 15009 | 6 |
| Video 11 (NON) | 6745 | 0 | 6639 | 106 |
| Video 12 (NON) | 14949 | 0 | 14943 | 6 |
| Video 13 (NON) | 14891 | 0 | 14875 | 16 |
| Video 14 (NON) | 14975 | 0 | 14965 | 10 |
| Video 15 (NON) | 4402 | 0 | 4387 | 15 |
| Video 16 (NON) | 13454 | 0 | 13448 | 6 |

Videos 1 to 8 contain smoke and fire flame and Videos 9 to 16 contain non-fire (factory and office) scenes. Video 3, Video 7, and non-fire Video included a number of false positive frames. Especially, Video 3 showed the same number of true positive frames and false positives. It means that each frame has False Positive object in the images. In Table 1, Ground Truth represents the total number of frames in the video, True Positive (TP) indicates when a fire flame and smoke is detected as fire flame and smoke. True Negative (TN) indicates that non-fire objects are not detected as fire flame and smoke. False Positive (FP) is a case where non-fire objects are detected as a fire. NON signifies a non-fire video and F/S signifies a fire flame and smoke video. In Table 1, F/S means including fire and smoke frames and NON means without fire and smoke frames.

In the case of Videos, they is not generated in a continuous frame. Since the video is 30 fps, it can be sufficiently compensated. However, in the case of Video 3, Video 7, and non-fire video, the alarm continues to ring and the stress of the worker becomes higher. In order to reduce false positives generated in False Positive Videos, we use the following characteristics. The first is a global check. We checked the motion characteristics before performing deep learning using mean square error (8) and three frame differences (9) [55]. Since there is motion when a fire occurs, if a block of moving pixels is generated, it is registered as a fire candidate state. If the fire candidate frame status is True, a deep learning process is performed, as shown Figure 1.

$$S_k = SSIM(f_i, f_j), \ M_k = MSE(f_i, f_j), \ A_k = diff(f_i, f_j) \tag{9}$$

$$FS_G = \begin{cases} 1 & if \ S_k < th1, \ M_k < th2, \ A_k < th3 \\ 0 & else \end{cases} \tag{10}$$

where FSG is global decision parameter.

The second is a local check for the detected area (bounding box) by deep learning. If there is a trained class in the input frame image, a bounding box is created and stored as a local area of interest. The next step is to verify the local area of interest again. In this paper, we determine the final fire region using the color histogram *H*, *SSIM* index, and mean square error (*MSE*), coefficient variant, and wavelet transform with other frames as the following equation:

$$
\begin{aligned}
F_L &= \begin{cases} 1 & if\ M_k < fth1,\ A_k < fth2,\ H_{sum\_F} < fth3,\ WE_k < fth4 \\ 0 & else \end{cases} \\
WE_k &= \sqrt{FWV^2 + C\_R\_HH^2 \times (R\_H_{sum} + Y\_H_{sum})} \\
FWV &= \sqrt{C\_R\_HH^2 \times CV + C\_Y\_HH^2 \times CV}
\end{aligned}
\tag{11}
$$

where *k* means frames, from *fth1* to *fth4* are threshold value by experiment. C_R_HH and C_Y_HH is the wavelet transform coefficient HH for RGB and *YCbCr* color. Moreover, $R\_H_{sum}$ and $C\_H_{sum}$ are the result of R color and Y color histogram for the local region. We compared the local region (bounding box area) of interest using the three frame difference algorithm (first, middle, and last frames) from the stored 10 frame images.

The final smoke region, in common with fire detection, we also adapted same sequence as the following equation:

$$
S_L = \begin{cases} 1 & if\ M_k > sth1,\ A_k > sth2,\ H_{sum\_F} > sth3,\ WE_k < sth4 \\ 0 & else \end{cases}
\tag{12}
$$

This paper added the following conditions to remove false positives:

$$
\begin{aligned}
SD1 &= \sqrt{C\_Y\_HH^2 \times FWV} \\
SD2 &= SD1 \times FWV \\
SD3 &= \{(CV_S + CV_F)/2\} \times SD2 \\
SD4 &= CV_S \times C\_Y\_HL\_LH \\
C\_Y\_HL\_LH &= \sqrt{C\_Y\_HL^2 \times FWV + C\_Y\_LH^2 \times FWV} \\
S_{SD} &= \begin{cases} 1 & if\ SD1 > sth5,\ SD2 > sth6,\ SD3 > sth7,\ SD4 < sth8 \\ 0 & else \end{cases}
\end{aligned}
\tag{13}
$$

where CVS and CVF are the coefficient variance of local smoke and fire region, respectively. In this paper, it is regarded as a fire if FD is satisfied as shown in the following equation:

$$
FS_G = \begin{cases} 1 & if\ F_L > 0,\ S_L > 0,\ F_{SD} > 0 \\ 0 & else \end{cases}
\tag{14}
$$

We described the result of the experiment applying the proposed algorithms in Table 2.

Table 2 shows the experimental results using the proposed algorithm. In the Videos, the false positive rate dropped to 0% and the fire detection of Video 1 to Video 6 persisted. Even though the Video 3 and Video 4 missed a few fire images, it has no problem because it is not continuously generated and the alarm system has no problem sending a warning signal to operator if it misses one or two frames. As shown in Table 2, the proposed algorithm using color histogram, wavelet transform, and coefficient variant was able to eliminate false positives (similar shape and color objects, sun and light reflection, moving objects, etc.) shown in Figure 8b,c. The results of the proposed algorithm using color histogram performance, high frequency components of wavelet transform, which is background discrimination of smoke and fire flame, and coefficient variant coefficients showed higher ratio of false alarm removal than the traditional deep learning method. However, in the case of Video 7 and Video 8, we must seriously consider the case of the missing frames. Additionally, we tested other factory and office videos. It also marked zero false positive rate for the proposed method. The false positive rate for the additional 16 videos was 99%, and the image examples used in the video experiment are shown

in Figure 9. Figure 9a is office and factory videos and Figure 9b is fire and smoke videos. Since this involves a lot of movement, it is likely that it has affected the frames missing in Video 7 and Video 8.

**Table 2.** The results of video test using proposed algorithm.

| Videos | Ground Truth | True Positive | True Negative | False Positive |
|---|---|---|---|---|
| Video 1 (F/M) | 85 | 85 | 0 | 0 |
| Video 2 (F/M) | 102 | 102 | 0 | 0 |
| Video 3 (F/M) | 890 | 888 | 0 | 0 |
| Video 4 (F/M) | 1159 | 1158 | 0 | 0 |
| Video 5 (F/M) | 1477 | 1477 | 0 | 0 |
| Video 6 (F/M) | 1112 | 1112 | 0 | 0 |
| Video 7 (F/M) | 544 | 502 | 0 | 0 |
| Video 8 (F/M) | 1040 | 949 | 0 | 0 |
| Video 9 (NON) | 12112 | 0 | 12112 | 0 |
| Video 10 (NON) | 15015 | 0 | 15015 | 0 |
| Video 11 (NON) | 6745 | 0 | 6745 | 0 |
| Video 12 (NON) | 14949 | 0 | 14949 | 0 |
| Video 13 (NON) | 14891 | 0 | 14891 | 0 |
| Video 14 (NON) | 14975 | 0 | 14975 | 0 |
| Video 15 (NON) | 4402 | 0 | 4402 | 0 |
| Video 16 (NON) | 13454 | 0 | 13454 | 0 |

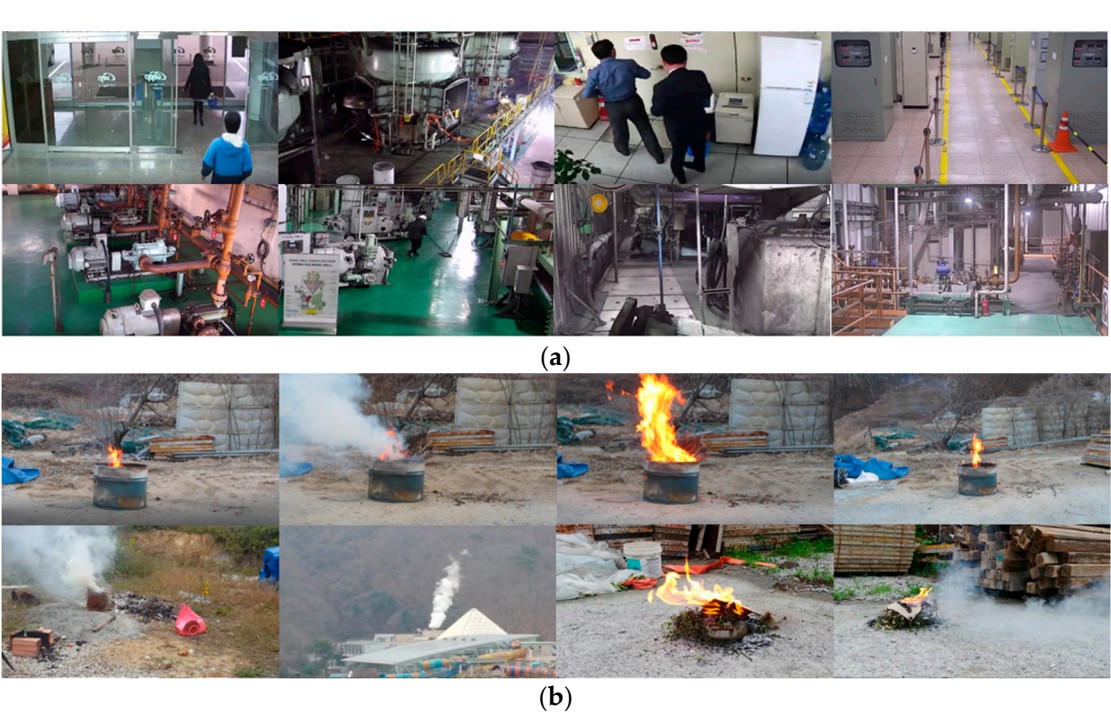

(**a**)

(**b**)

**Figure 9.** Experimental videos for proposed algorithm test: (**a**) factory and office videos and (**b**) fire and smoke videos.

## 5. Conclusions

Fires resulting from small sparks can cause terrible natural disasters that can lead to both economic losses and the loss of human lives. In this paper, we describe a new fire flame and smoke detection method to remove false positive detection using spatial and temporal features based on deep learning from surveillance cameras. In general, a deep learning method using the shape of an object frequently generate false positives, where general object is detected as the fire or smoke. To solve this problem, first, we used motion detection using the three frame difference algorithm as the global information.

We then applied the frame similarity using SSIM and MSE. Second, we adapted the Faster R-CNN algorithm to find smoke and fire candidate region for the detected frame. Third, we determined the final fire flame and smoke area using the spatial and temporal features; wavelet transform, coefficient of variation, color histogram, frame similarity, and MSE for the candidate region. Experiments have shown that the probability of false positives in the proposed algorithm is significantly lower than that of conventional deep learning method.

For future work, it is necessary to study the analysis for the moving videos and the experiment using the correlation of the frame and the deep learning model to further reduce false positives and missing fire and smoke frames.

**Author Contributions:** We provide our contribution as follow, conceptualization, Y.L. and J.S.; methodology, Y.L. and J.S.; software, Y.L.; validation, Y.L.; formal analysis, Y.L.; investigation, Y.L.; resources, Y.L.; data curation, Y.L.; writing—original draft preparation, Y.L.; writing—review and editing, Y.L. and J.S.; visualization, Y.L.; supervision, J.S.; project administration, J.S.; funding acquisition, J.S.

**Funding:** This research was supported by the MSIP (Ministry of Science, ICT & Future Planning), Korea, under the National Program for Excellence in SW) (IITP-2019-0-01113) supervised by the IITP (Institute for Information & communications Technology Planning & Evaluation).

**Conflicts of Interest:** The funders had no role in the design of the study; in the collection, analyses, or interpretation of data; in the writing of the manuscript, or in the decision to publish the results.

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
