# Peer review of "False Positive Decremented Research for Fire and Smoke Detection in Surveillance Camera using Spatial and Temporal Features Based on Deep Learning"

_electronics, doi:10.3390/electronics8101167_

Round 1
Reviewer 1 Report
General Comments:
The text is very confusing, the language is not well written and must be corrected by a native. There are many syntax errors, examples of these problems can be found at:
Line 12:
“…precious lives. And vision base has many difficulties…”
Line 70-73:
“Conventional flame detection methods are a method using…”, “models and etc., wavelet transform”
Line 104:
“…but sometimes smoke is absolutely transparent and sometimes fails”
Line 122:
“Frizzi et al [41]. researched the Convolution Neural Network (CNN) based smoke and flame detection..”
Line 128:
“Zhang et al. [46] researched forest fire detection utilize fire patches detection using two joined..”
The paper is not well distributed, methods and results are mixed. The introduction is especially confusing, it is a list of the intentions of previous work, but without going into detail of the results or limitations they present. Most acronyms are not defined in the text (CCD, RGB, HSV, HIS, CIEL*a*b, LPB, LBPV, YOLO, R-CNN, SVM…). Figure one is in the introduction and is not mentioned in the text until the experimental results in section 7.
Section 2:
Your intention is to explain Deep Learning basis, the methods you follow or the methodology that you develop?
This comment can be applied along sections 3, 4, 5 and 6.
Most of the bibliography corresponds to results from conferences, symposiums and proceedings.
Specific comments:
Line 41:
What do you mean with infrared rays? Infrared spectrum? Infrared spectrum is too large, which part of the infrared spectrum? Thermal?...
I do not know any ultraviolet sensor designed to detect flames, but for sure is not described in references [2,3].
Line 166:
Figure 2 is full of acronyms that have not yet explained in the text (RPN, ROIP, Y, V>TH). This figure does not explain efficiently the R-CNN system flow, for example, section 2.1 is not present in the flowchart.
Line 174:
“This paper has used a variety of fire flame…” The paper did not use anything, the authors use.
Line 184:
“Faster R-CNN is a method of applying a new method called Region Proposal Network…” This makes no sense. Do you develop this method (RPN)? If not provide reference.
Line 193:
“RPN extracts 256 or 512 features from input image” why 256 or 512? what do you mean with features? Why not 257?
Lines 229-231:
C is an image, but with only one value? What is the meaning of CL, CH? And the τ1, τ2…? Then C is an average image?
Line 295:
What do you mean with “whole frame and local frame”?
Lines 297-299:
This is methodology not results.
Line 296:
What is a Onvif camera? Is it a camera? Then you use a mobile phone camera, from iphone? from Nokia? What year? Sensor size? Megapixels? HDR? Focal length?
Line 301-302:
What do you mean with “basically”? there are other software? Opencv as a Python library? As a commercial software? Python 2.7, 3.5, other?
Line 303-307:
This is methodology not results.
What do you mean with fire and non fire video? This mean that there are not videos with a start of a fire, I mean with part of the video without fire or smoke and part of the video with fire and/or smoke. Do your method needs the whole frames of the video in order to provide detection? This working method would make real-time detection impossible.
Author Response
I attached response letter.

Reviewer 2 Report
This paper designs a detection algorithm which reduces the false positive rate of fire and smoke detection. Their detection algorithm based on Faster R-CNN. Then they design a set of algorithms to remove the false positive. Here are my minor questions:
One of the contribution of this work is the design of the training dataset. If this dataset is open source? Also if this algorithm can work on other existing datasets? Can you show how many false positive frames can be removed by each algorithm individually? Currently it is not clear which techniques solves which kind of false positive, i.e. the effectiveness of each of them.Author Response
I attached response letter.

Round 2
Reviewer 1 Report
Specific comments:
Lines 155-159:
You explain 5 sections, but in the paper there are 8 main titles (1. Introduction, 2. Deep Learning (Faster R-CNN), 3. Structural Similarity, 4. RGB Color Histogram, 5. Coefficient of Variation (CV)…)
Please, clarify which section correspond with each main title.
Lines 275-277:
There are values with many decimal values, please round to only significant ones.
Lines 314-315:
This corresponds to the lines 301-302 in the first version of the paper. “The Python, Tensorflow, and Opencv were basically used in this paper.” This sentence has not been removed as you respond on the cover letter; the word “basically” should be removed.
Lines 427-428:
“In Figure 9, upper two rows are office and factory videos and bottom two rows are fire and smoke videos.” It seems that office and factory videos are the bottom two rows and fire and smoke correspond to the firs two rows. Clarify by including label letters on each video and referring the comments to the corresponding labelled video frames.
Lines 436-437:
“A fire caused by a small spark is can lead to a terrible nature disaster which can lead to loss in one’s fortune and human lives.” “Is can lead” is not correct; in addition replace “nature” by natural. Although the English language has been improved in this version, please review the entire document.
Line 448:
“As a study on the future, it is necessary to study” too many “study”, consider replace by something like “as future research, it will be necessary to study…”
Author Response
Thank you for your comments.

Reviewer 2 Report
The authors made substantial efforts to revise the manuscript. I agree their publication. One recommendation is to repeat the reviewers' questions in the reply letter to improve readability and save reviewers' time. The English is still a big issue which should be carefully handled by authors before its official publication.
Author Response
Thank you for your comments
